# Developmental Trajectories of Cyber-Aggression among Early Adolescents in Canada: The Impact of Aggression, Gender, and Time Spent Online

**DOI:** 10.3390/ijerph21040429

**Published:** 2024-04-01

**Authors:** Bowen Xiao, Natasha Parent, Takara Bond, Johanna Sam, Jennifer Shapka

**Affiliations:** Department of Educational and Counselling Psychology, University of British Columbia, Vancouver, BC V6T1Z4, Canada; natashaparent@gmail.com (N.P.); takara.bond@alumni.ubc.ca (T.B.); johanna.sam@ubc.ca (J.S.); jennifer.shapka@ubc.ca (J.S.)

**Keywords:** cyber-aggression, developmental trajectories, overt aggression, gender

## Abstract

The objective of the present study was to examine developmental trajectories of cyber-aggression in early adolescence, as well as their relationship with predictive factors related to cyber-aggression (e.g., overt aggression, gender, and time spent online). Participants were 384 adolescents from the Lower Mainland of British Columbia, Canada who were in grade six and grade seven at Time 1 of the study (192 boys, Mage = 13.62 years, SD = 0.74 year). Three years of longitudinal data on cyber-aggression, overt aggression, and time spent online were collected via online self-report questionnaires. Findings indicated three different trajectories of cyber-aggression: (a) a low-increasing (85.7% of the sample), (b) a stable trajectory (9.3% of the sample), and (c) a high-decreasing trajectory (4.9% of the sample). Adolescents who reported higher scores on overt aggression and spent more time online were more likely to be in the stable or high-decreasing groups. These findings highlight the importance of studying subgroups regarding the developmental course of cyber-aggression in early adolescence. The implications of present study findings give insight into gender differences and overt aggression among youth to inform cyber-aggression intervention and prevention.

## 1. Introduction

With increasing communications taking place online, cyber-aggression has emerged as a major public health concern affecting children and adolescents [1,2,3]. In fact, data from Statistics Canada indicate that 35% of Canadian children reported being a victim of cyberbullying in 2022. Similar to traditional forms of bullying, cyber-aggression involves harassing, insulting, physically threatening, and/or humiliating others via the internet or other forms of electronic media [3]. Importantly, accumulating and converging evidence indicates that cyber-aggression is concurrently and predictively associated with adjustment difficulties [4,5]. For example, researchers) found that cyber-aggression uniquely predicted academic problems (e.g., greater absenteeism and poor grades in school) as well as increased depression, anxiety, and poor self-esteem over and above traditional bullying [6,7,8]. Unsurprisingly, given the prominent role of technology in our lives, research on adolescents’ cyber-aggression has grown substantially [9,10,11,12]. Furthermore, previous research on cyber-aggression has predominantly focused on examining the role of environmental factors as potential risk factors [13,14]. While these studies have provided valuable insights, limited attention has been given to investigating the influence of individual factors, such as aggression, and behavioral factors, such as time spent online, as risk factors for predicting trajectories of cyber-aggression, particularly among Canadian adolescents. Accordingly, the goal of the present study was to investigate the developmental trajectories of cyber-aggression among Canadian adolescents.

### 1.1. Cyber-Aggression during Adolescence

Cyberbullying refers to aggression that occurs through technology such as mobile phones or the Internet [15] and is a common and harmful experience among adolescents. Indeed, previous studies have shown that victimization rates tend to range between 17.4% and 28.3% [16], while perpetration rates range widely from 5% to 35% across studies, and dual involvement in both cyberbullying and cybervictimization ranges from 3% to 14% in North America [17]. In a school-based study of Canadian youth (*N* = 2186) in middle and high school, close to half (49.5%) of the participants indicated that they experienced cyber-aggression [18]. It is crucial to consider the traditional criteria of bullying, which typically involves repeated attacks (often defined as occurring 3–4 times a month) and a power imbalance. However, in the digital realm, the frequency and power dynamics can manifest differently. For instance, a single online act can have a widespread and lasting impact, challenging the traditional frequency criterion of bullying [19]. While being cyberbullied has been linked with adjustment difficulties and behavioural problems [4,20,21,22], studies have found that those doing the cyber-aggression are also at risk for increased harms. For example, cyber-aggression towards others has been linked with externalizing problems, such as hyperactivity, conduct problems, substance use, and peer problems [23,24,25,26], internalizing problems, such as stress, depression, and anxiety [27], and increased tobacco, alcohol, and drug use [28]. Moreover, adolescents who experience cyber-aggression or cybervictimization were more likely to report suicidal thoughts or suicide attempts [29]—thus, there is an urgent need to address and reduce cyberbullying behaviours among this young demographic.

### 1.2. The Trajectories of Cyber-Aggression in Adolescence

As adolescents develop the capacity to use new technologies, they spend an increasing amount of time engaged in online activities, such as social networking, instant messaging, gaming, and consuming content [30], as compared to younger children. Indeed, the use of technology in adolescence provides additional opportunities for socialization [31], which can directly contribute to the attainment of adolescents’ developmental social goals [32]. Although there are many benefits to using the internet, it also places adolescents at increased risk for experiencing cyber-aggression [33]. Indeed, previous studies have consistently demonstrated that the average developmental trajectory of cyberbullying and cybervictimization increases during adolescents [34]. Specifically, studies have found that as adolescents grow older, their likelihood of engaging in cyber-aggression behavior increases [35,36], as well as the likelihood that they will be a victim of cyber-aggression [37].

Furthermore, while average developmental trajectories provide an overview of the typical cyber-aggression patterns among children and adolescents, it is important to recognize that not all individuals engage in cyber-aggression behavior in the same way over time [38,39]. Indeed, researchers in this area have found that there exist distinct developmental trajectories of cyber-aggression which can be observed in longitudinal analyses. For example, researchers examined the developmental trajectories of both traditional bullying and cyberbullying perpetration using a four-wave longitudinal study among early adolescents, 10–14 years old, in Belgium [17]. They identified four distinct groups within their analysis: (1) nonstop traditional bullies (9.8% of the sample), (2) traditional and cyber bullies with decreasing perpetration (7% of the sample), (3) traditional and cyber bullies with increasing perpetration (2.5% of the sample), and (4) non-involved (80.7% of the sample) [26]. They further found that social intelligence was associated with less nonstop traditional bullying. Furthermore, several studies conducted by researchers have identified three distinct subgroups of trajectories related to cyberbullying among South Korean adolescents. These subgroups include: (1) a normative trajectory group/low risk group/non-involved group, which comprised the majority of the sample across all four studies, (2) an increasing and late-peak group/high risk increasing group/bullying group, and (3) an early onset and decreasing group/chronic group/transient group [28,39,40,41]. Furthermore, they identified that the most significant explanatory factors for both cyberbullying and traditional bullying included personal factors, such as low self-control and self-esteem, environmental factors, such as limited support from parents and friends and reduced parental supervision, and behavioral factors, such as prior perpetration experience and access to cyberspace. While these findings are instrumental to uncovering the developmental trajectories of cyber-aggression in adolescence, and thus have important implications for the development of intervention and prevention efforts targeting cyber-aggression perpetrators, there has yet to be a study exploring these trajectories among Canadian adolescents. While we anticipated similar cyber-aggression trajectories in Canadian adolescents as observed in other countries, this contextualization offers a more nuanced insight into the unique manifestations and mechanisms of cyberbullying within the Canadian cultural and societal context. In addition, conducting a comparative analysis of cyberbullying trajectories in Canada adds to the broader literature by expanding the range of countries and populations studied. Therefore, this study aimed to fill this gap through investigating the possibility of distinguishing different subgroups of adolescents’ cyber-aggression trajectories in Canada, with a specific focus on the role of overt aggression, gender, and time spent online.

### 1.3. Risk Factors of Cyber-Aggression

Bronfenbrenner’s ecological systems theory posits that an individual’s development is affected by various layers of environmental systems, ranging from immediate settings like family and school to broader societal influences [42]. In the context of cyber-aggression, this theory helps us to understand how different environmental layers interact with individual characteristics (like gender and aggression) and behaviors (such as time spent online) to influence the emergence and progression of cyber-aggressive behaviors [43,44,45]. While previous research has often focused on the broader environmental factors, such as peer relations and parenting, as risk factors for cyber-aggression [46], less attention has been paid to how these environmental contexts interact with individual and behavioral factors to shape cyber-aggression trajectories, particularly among Canadian adolescents. Therefore, in the present study, our aim was to fill this research gap by specifically examining individual and behavioral factors as potential risk factors for cyber-aggression trajectories among Canadian adolescents. By focusing on these factors, we sought to gain a deeper understanding of how they contribute to the developmental pathways of cyber-aggression within the Canadian context.

### 1.4. Trajectories of Cyber-Aggression and Relations with Overt Aggression

While practically inherent to traditional forms of bullying, overt aggression, which refers to actions such as hitting, kicking, and punching with the intent of causing bodily harm to the target [47,48,49], has also been consistently linked with cyberbullying behaviour [36]. For example, a study by found that traditional bullying (often characterized by overt aggression) significantly predicted cyberbullying among adolescents [50], and a study by found that aggressive behaviours in school were highly associated with cyber-aggression perpetration [51]. Indeed, researchers have suggested that cyber-aggression is a specific type of aggression [52,53] and therefore, adolescents who engage in overt aggression are more likely to bully others online as well. Moreover, researchers have further argued that the Internet may provide a means of extending traditional bullying that takes place at school [5], by allowing for interpersonal relations at school, including bullying, to be extended after school time. This has been supported by researcher who discovered that 64% of adolescents reported that their personal experience with bullying began at school, often offline, and then continued online once they got home [54]. As such, it is reasonable to speculate that adolescents who exhibit overt aggression are more likely to engage in cyber-aggression overtime.

### 1.5. Trajectories of Cyber-Aggression and Relations with Time Spent Online

Time spent online has been consistently associated with cyber-aggression [55]. Specifically, it has been theorized that spending time online increases the likelihood of being exposed to violent content, including cyber-aggression, and that adolescents may come to accept, and even replicate, it [56,57,58,59,60]. In fact, previous studies have identified the frequency of internet use as a key contributor to adolescents’ cyber-aggression [61,62,63,64]. For example, researchers found that adolescents who cyberbully spend significantly more time online than their peers [55]. More recently, Yudes et al. (2020) found that time spent online significantly predicted cyber-aggression behaviour among Spanish adolescents [65]. Thus, findings from this previous work provide a strong foundation from which we expect time spent online to be a significant factor contributing to cyber-aggression in our study.

### 1.6. Gender Differences in Cyber-Aggression

Previous work has been inconsistent in identifying gender differences in cyber-aggression [66,67,68,69] For example, a study by Kowalsky and Khurana et al. (2015) found that girls were more likely to engage in cyber-aggression than boys [50]. The authors explained that girls may be more likely to engage in this type of aggression due to the indirect nature of cyber-aggression. In addition, girls may face different peer pressure and group dynamics that lead to engagement in cyber-aggression as a means of fitting in or gaining social approval. However, studies by Lee and Shin (2017) and by Slonje and Smith (2008) found that boys reported higher levels of cyberbullying than girls [70], and many other studies have found no gender differences [71]. For example, Hellström (2012) found that there were no statistically significant gender variations, with 0.7% of boys and 0.8% of girls being involved in cyberbullying behaviors [45]. Due to this reported inconsistency, the present study examined gender differences in adolescents’ cyber-aggression trajectories.

### 1.7. The Present Study

The current study used a cohort-sequential three-year longitudinal design to explore the trajectories of cyber-aggression among young Canadian adolescents. The primary aim of this study was to identify distinct latent trajectories of cyber-aggression across the period of early adolescence. Based on findings from previous work, we postulated that most adolescents would demonstrate increasing trajectories of cyber-aggression and that the remaining adolescents would exhibit either decreasing or stable trajectories of aggression. The second aim of this study was to determine whether cyber-aggression trajectories could be discriminated based on levels of overt aggression, gender, and time spent online. Considering findings from previous work, we hypothesized that adolescents higher in overt aggression, as well as those who spent more time online, would have a higher likelihood of displaying a stable or increasing trajectory of cyber-aggression than adolescents who were lower on overt aggression and spent less time online.

## 2. Methods

### 2.1. Procedure

The present study obtained ethical approval from the UBC Behavioural Research Ethics Board. Participants were enlisted from sixteen public schools in Southern British Columbia, Canada. Written consent from both the participants and their parents was secured via the schools, where personal contact details were also gathered. The study spanned three years, with data collection occurring at three points: Time 1 (T1), Time 2 (T2), and Time 3 (T3). During T1 and T2, graduate research assistants facilitated the completion of an online self-report questionnaire by the adolescents within school premises. The participants provided information about their sociodemographic background, experiences with cyber-aggression, and online activity duration, utilizing iPads provided by the research team. By T3, as the adolescents moved to secondary school, they were reached out to through texts, emails, and calls to reconfirm their participation. Following this, they received a link and personal login credentials to complete the online questionnaire at their own pace.

### 2.2. Participants

In the first wave (T1, 2014), participants were 384 students from grade six and grade seven (192 boys; *M_age_* = 13.62 years, *SD* = 0.74 year). There were *n =* 230 sixth-grade students (128 boys, 104 girls; *M_age_* = 12.34 years, *SD* = 0.43 year) and *n =* 154 seventh-grade students (64 boys, 81 girls; *M_age_* = 13.65 years, *SD* = 0.45 year). Approximately 74.8% of the students were born in Canada, and 51.2% of them were of Asian descent. At T2 and T3, 212 and 204 participants provided effective data, respectively. All data were used in the analysis. Little’s (1988) MCAR test was conducted to investigate the impact of missing data. Results showed that the missing data pattern was not systematic (χ^2^ (95) = 117, *p* = 0.07). In addition, there was no significant difference for adolescents who dropped out of the study on all variables (*p* > 0.05). Thus, we used the Full Information Maximum Likelihood (FIML) to estimate missing data in Mplus 7.2 [72].

### 2.3. Measures

#### 2.3.1. Sociodemographic Information

Participants self-reported their age, gender, and ethnicity.

#### 2.3.2. Cyber-Aggression

The *Cyber-aggression Scale*, developed by Shapka and Maghsoudi (2017), comprises six items assessing cyber-aggression behaviors, such as posting harmful content online or sending hurtful messages electronically [73]. It employs a scale ranging from 0 (“never occurred”) to 4 (“occurs daily”). Previously utilized in Canadian research, this scale has demonstrated reliable psychometric qualities [73]. In the current study, it exhibited high internal consistency, with a Cronbach’s alpha of 0.89.

#### 2.3.3. Overt Aggression

The *Form and Function Aggression Scale-overt aggression subscale* [73] is a six-item measure of overt aggression (e.g., “I’m the kind of person who often fights with others”) that uses a four-point Likert-type scale (0 = “not at all true about me,” 1 = “sometimes true about me,” 2 = “often true about me,” 3 = “always true about me”). This measure has been successfully used in previous work and has been shown to have sound psychometric properties [74]. Internal consistency in the present sample was α = 0.93.

#### 2.3.4. Average Hours Online

The average daily online time was determined by merging responses from two questions regarding the typical hours spent online on weekdays and weekends (e.g., “On a typical weekday (Monday to Friday) when you use the internet, how many hours do you spend online?”). The average weekday online duration was 2.56 h (SD = 3.32), while for weekends, it was 3.48 h (SD = 3.81). To calculate the overall daily average, we summed the total weekly hours and divided by seven. This method of measurement has been effectively employed in prior Canadian studies [73].

### 2.4. Statistical Analytic Strategy

Initially, to verify the consistent measurement of the optimally fitting baseline model across three time points, we executed a multi-group confirmatory factor analysis (CFA). We assessed configural invariance through the previously mentioned model fit indices (i.e., χ^2^, CFI, TLI, RMSEA). Metric invariance was deemed established if there was no significant worsening in model fit compared to the configural invariance (ΔCFI < 0.01 and ΔRMSEA < 0.015). Scalar invariance was affirmed when the model fit was comparably stable against the metric invariance criteria (ΔCFI < 0.01 and ΔRMSEA < 0.015), following Chen’s (2007) guidelines [73,75].

We applied Latent Growth Curve Modeling (LGCM) to examine the overall trend of cyber-aggression in adolescents. Our data were structured using an accelerated longitudinal design based on the participants’ grade levels [11]. To determine the model’s absolute fit, we employed indices such as the Comparative Fit Index (CFI), Tucker-Lewis Index (TLI), Root Mean Square Error of Approximation (RMSEA), Standardized Root Mean Square Residual (SRMR), and the χ^2^ significance test [76,77]. Next, to investigate specific trajectory subgroups, we utilized Growth Mixture Modeling (GMM), assessing model fit through indicators like Akaike Information Criteria (AIC), Bayesian Information Criteria (BIC), sample-size Adjusted BIC (ABIC), entropy, Lo Mendell Rubin Test (LMR), Bootstrapped LRT (BLRT), and practical utility [72]. Subsequently, we employed the Mplus “three-step approach” [78], using multinomial logistic regressions to regress class membership on predictors such as gender, overt aggression, and average online hours, all measured at the baseline. The results for each of the three classes are detailed. All statistical analyses were conducted using Mplus 7.2 [72].

## 3. Results

### 3.1. Preliminary Analysis

Means, standard deviations, and correlations among gender, cyber-aggression, overt aggression, and time spent online are presented in Table 1. Moreover, the time invariance of psychometric measurement was tested for cyber-aggression. The fully constrained model, which had equal loadings and intercepts across time, had adequate levels of fit (except for ∆CFIs), CFI = 0.92, TLI = 0.91, SRMR = 0.07, ∆CFIs = −0.12. These results indicated that the measurement of cyber-aggression was weak invariant across time in our study. Furthermore, the results from LGCM showed that the model provided good fit for the data (model fits: χ^2^(1, *N* = 349) = 12.98, *p* = 0.011; CFI = 0.94; TLI = 0.92; RMSEA = 0.06; SRMR = 0.04). The general developmental trajectory of cyber-aggression was increasing from grade six to grade nine (*M_intercept_* = 0.22, *p* < 0.001; *M_slope_* = 0.05, *p* = 0.002; Variance of intercept = 0.19, *p* < 0.001; Variance of the slope = 0.04, *p* < 0.001).

### 3.2. Trajectories of Cyber-Aggression

The findings shown in Table 2 reveal that the three-class model was superior to both the two-class and four-class models in terms of fit. This is evidenced by the lower AIC, BIC, and ABIC scores for the three-class model, signifying a better model fit. Furthermore, the increased entropy value for the three-class model indicates more accurate classification. The three separate cyber-aggression trajectories identified are presented along with their respective estimated mean trajectories and individual value estimates in Figure 1. The trajectories contained (a) a low-increasing trajectory (298 students, 85.7% of the sample, intercept = 0.06, SE = 0.01, t = 4.61, *p* < 0.001; slope = 0.09 SE = 0.02, t = 6.10, *p* < 0.001): this trajectory started with low level of cyber-aggression in grade six and kept increasing; (b) a stable trajectory (32 students, 9.3% of the sample, intercept = 0.67, SE = 0.09, t = 7.52, *p* < 0.001; slope = 0.17, SE = 0.11, t = 1.52, *p* = 0.13): this trajectory began with a moderate level of cyber-aggression in grade six which remained stable over time; and (c) a high-decreasing trajectory (17 students, 4.9% of the sample, intercept = 1.91, SE = 0.07, t = 25.66, *p* < 0.001; linear slope = −0.62, SE = 0.07, t = −8.42, *p* < 0.001): this trajectory started with a high level of cyber-aggression in grade six and kept decreasing.

### 3.3. Conditional GMM with Covariates Predicting Class Membership

We used average hours online, overt aggression, and gender as predictors in the optimal model. Our results indicated that the high-decreasing group had higher levels of overt aggression and average hours online than the low-increasing group. The stable group also had higher levels of overt aggression and average hours online than the low-increasing group. Gender did not significantly predict the latent class membership (see Table 3).

## 4. Discussion

The present study examined trajectories of cyber-aggression in a sample of Canadian adolescents, and explored the role of overt aggression, gender, and time spent online in predicting each trajectory. We identified three distinct trajectories of cyber-aggression in our analysis: (a) a low-increasing trajectory (85.7% of the sample), (b) a stable trajectory (9.3% of the sample), and (c) a high-decreasing trajectory (4.9% of the sample). Moreover, our findings indicated that adolescents who reported higher scores on overt aggression and spent more time online were more likely to be in the stable and moderate-increasing trajectory groups. In other words, early overt aggression and time spent online were significant risk factors predicting later cyber-aggression.

### 4.1. Trajectories of Cyber-Aggression in Adolescence

Overall, our findings contribute to the growing body of literature indicating that most adolescents report low levels of cyber-aggression [22,79]. For example, Pabian and Vandebosch (2016) found that most German adolescents fell in the (cyberbullying) non-involved group (80.7% of the sample) [26]. Similarly, Cho and Glassner (2020) found that most Korean adolescents (14–19 years old) belonged to the normative trajectory group (i.e., low levels of cyberbullying; 91.3% of the sample) [41]. Moreover, consistent with previous studies, our results also indicated that adolescents’ self-reported cyber-aggression increased over three years. This is consistent with the results of a meta-analysis by Guo (2016) which found that cyberbullying increased with age, and findings by Monks et al. (2012) that older children reported higher levels of cyberbullying than younger children [34]. These findings may be partly explained by the increased access to the Internet and social media in adolescence [10], which may provide greater opportunities to engage in cyber-aggression [33].

Although most adolescents in our study were in the cyber-aggression increasing trajectory group (i.e., low levels of cyberbullying in grade six), we did find two other subgroups of trajectories of cyber-aggression (i.e., the stable trajectory group, 9.3% of the sample, and the high-decreasing trajectory group, 4.9% of the sample). Again, consistent with previous studies, there were a small number of adolescents who either began with a high level of cyber-aggression or whose cyber-aggression remained stable over time [41]. For the high-decreasing group, it is possible that the transition from elementary to secondary school often brings about changes in peer groups and social dynamics. Students may distance themselves from earlier behavior patterns as they seek to establish new identities and social standings in the more diverse and complex social environments of secondary schools. For the stable group, some students might have personality traits or personal circumstances that make them more prone to engage in cyber-aggression consistently. For example, individuals who are more aggressive or those who have less supportive social networks might continue to engage in cyber-aggression activities. In addition, the convergence of cyber-aggression levels between the high-decreasing group and the stable group by grade nine could reveal important insights into the dynamics of cyber-aggression behavior over time. Specifically, the convergence of cyber-aggression levels between the high-decreasing and stable groups by grade nine could be due to several factors, including developmental maturation, where adolescents develop better self-regulation and empathy, leading to decreased aggressive online behaviors. School transitions may also play a role, as entering high school often reshuffles peer groups and social dynamics, potentially normalizing behaviors across different groups. It is also possible that the convergence of cyber-aggression levels between the high-decreasing and stable groups may be attributed to “regression to the mean,” suggesting that the initial differences were due to their starting points rather than inherent differences in cyberbullying behavior. Essentially, both groups might naturally align over time as the initially higher rates in the high-decreasing group decrease to match the stable level observed in the other group.

Notably, previous work indicates that these adolescents may be especially vulnerable to adjustment issues and to causing harm to others [25]. Consequently, our findings suggest that younger age groups may be especially receptive to proactive cyber-aggression prevention initiatives, aimed at curbing the initial emergence of such behaviors. On the other hand, older age groups may benefit more from intervention strategies, since the data suggest a slight increase in cyber-aggression involvement over time within this demographic. The observed age-related disparities in cyber-aggression trajectories highlight the importance of customizing anti-cyber-aggression programs to align with the developmental stages of specific age groups within educational settings. This tailored approach can maximize the effectiveness of interventions by addressing the unique challenges and needs of students at various points of their educational journey.

### 4.2. Trajectories of Cyber-Aggression and Relations with Overt Aggression

Our results are congruent with previous studies indicating that adolescents higher in overt aggression are more likely to engage in cyber-aggression [36,71]. Indeed, in line with the work of Hemphill et al. (2015) our findings suggest that aggressive behaviour may be a significant longitudinal risk factor of cyber-aggression [45]. As suggested by Ang et al. (2014), we posit that this is because cyber-aggression is a specific type of aggression—and thus, adolescents who engage in overt aggression may be more likely to engage in this type of cyberaggression as well [52]. The current study findings highlight the influence of overt aggression on cyber-aggression behaviours provides a nuanced understanding of aggression during adolescence.

This study’s significance becomes apparent when we consider the identification of early overt aggression as a pivotal risk factor for later engagement in cyber-aggression. Indeed, this finding adds substantial depth to the body of research exploring the developmental pathways of cyber-aggression and carries vital implications for interventions and prevention efforts. What this suggests is that programs designed to combat cyber-aggression may yield their greatest impact when directed toward adolescents who exhibit signs of early overt aggression. In light of these results, educators and researchers should prioritize the early identification of children and young adolescents displaying overtly aggressive behaviors. By intervening at this crucial stage, the aim is to mitigate their risk of transitioning into more severe cyber-aggression involvement as they progress through their formative years. This tailored, proactive approach may be instrumental to reducing the overall prevalence of cyber-aggression among adolescents.

### 4.3. Trajectories of Cyber-Aggression and Relations with Time Spent Online

Consistent with our hypothesis, we found that average time spent online could also predict cyber-aggression trajectories. This finding supports that of previous studies which have shown that Internet usage is a key contributing factor to adolescents’ cyber-aggression behaviour [61]. For example, Erdur-Baker (2010) discovered that cyberbullying adolescents spend considerably more time online compared to their peers [55]. Similarly, Yudes et al. (2020) also demonstrated a significant correlation between time spent online and cyberbullying behavior among Spanish adolescents [65]. The internet provides easy access to a vast amount of information and online platforms where individuals can interact. This accessibility makes it more likely for people, including adolescents, to come across violent content or engage in cyber-aggression behaviors [56]. Adolescents who are frequently exposed to such behavior may develop acceptance for it and even imitate it [58]. Again, these findings have important practice implications. For example, it underscores the importance of active parental and teacher involvement in the online activities of children and students. That is, parents and educators should actively engage with, and guide, children in their online experiences. By helping support children and adolescents in navigating their relationships online, and offering guidance, adults can play a pivotal role in reducing the risk of cyber-aggression behaviors among adolescents. This proactively supportive approach is instrumental to creating a safer online environment for young individuals.

### 4.4. Gender Differences

Gender did not significantly predict the latent class membership. There remains a lack of consensus around the role of gender in predicting cyber-aggression behaviour [67]. For example, some studies have found boys report higher levels of cyber-aggression than girls [70]. Specifically, past longitudinal study findings indicate that being an adolescent boy with an awareness of online risks and school bullying involvement was associated with increased risk for cyber-aggression behaviours [25]. Furthermore, in a meta-analysis, Sun et al. (2016) reported that boys were more involved in cyberbullying, and that their rate of online bullying was comparable to what was seen in the face-to-face environment [80]. However, other works pointing to gender differences in types of aggression, whereby boys are theorized to be more likely to get involved in direct aggression (e.g., physical aggression) and girls to be more likely to get involved in indirect aggression (e.g., spread gossip) [81], suggest that girls may be more likely to engage in cyber-aggression g (e.g., spreading rumours online). Considering the varied results concerning gender’s impact on cyber-aggression trajectories, future studies should investigate the connection between gender and different forms of aggression, such as physical and relational aggression, and their effects on cyber-aggression.

### 4.5. Contributions, Limitations, and Future Directions

The results of the present study make a substantial contribution to our understanding of the developmental trajectories of cyber-aggression in young adolescence in a Western context. Specifically, we identified three distinct developmental trajectories (i.e., a low-increasing trajectory, a stable trajectory, and a high-decreasing trajectory) which were predicted by overt aggression and time spent online. Our findings have implications for prevention and intervention strategies for children and adolescents who are at risk of engaging in cyber-aggression. Notably, interventions for Canadian adolescents who exhibit a high degree of overt aggression and spend more time online may be particularly useful in helping reduce their chances of embarking on a stable or high-decreasing trajectory of cyber-aggression in early adolescence.

Despite this study’s contributions to the existing literature, some limitations should be acknowledged. First, the current study was conducted over a three-year period, in young adolescents from grade six and grade seven. Given the prevalence of cyber-aggression in later adolescence [41,82], and emerging adulthood [40], future studies should address longer-term trajectories of cyber-aggression behaviour to better understand how these trajectories play out across development. Moreover, the sample was limited to Canadian adolescents in British Colombia. Given that there are significant geographical differences in terms of social and cultural values within Canada, these results should be interpreted with caution when generalizing to other regions. Future studies should look into whether these findings are applicable to other parts of the globe. Finally, we only looked at how overt aggression, time spent online, and gender influenced cyber-aggression trajectories in this study. Given that previous research has shown that personality traits [83,84], parental supervision [40,85], as well as attitudes toward aggression (Burton et al., 2013) and Internet addiction [86], have a strong influence on cyber-aggression, future studies should further investigate the associations between these factors and cyber-aggression trajectories.

Even with these limitations considered, the current study highlights the risk factors of overt aggression and time spent online in predicting cyber-aggression in adolescence. That is, adolescents who are involved in offline aggressive behaviours and spend more time online may benefit from targeted prevention efforts to minimize future cyber-aggression behaviours. The present study findings suggest that developing age-appropriate intervention and prevention strategies may help protect youth against involvement in cyber-aggression. As youth move into middle and older adolescence, these cyber-aggression prevention and intervention should include aspects of their lives in both online and offline contexts.

## Figures and Tables

**Figure 1 ijerph-21-00429-f001:**
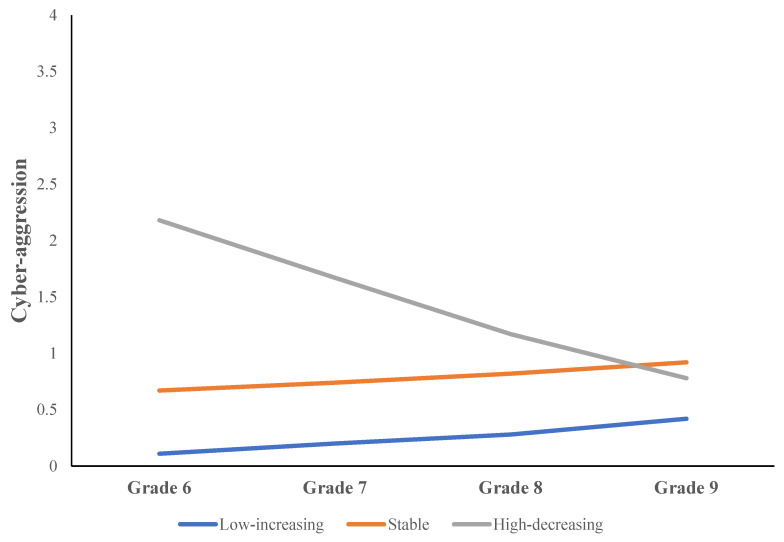
An exploration of cyber-aggression trajectory over time among adolescents.

**Table 1 ijerph-21-00429-t001:** Correlations, means, and standard deviations of gender, cyber-aggression, overt aggression, and spend time online.

	1	2	3	4	5	6
1. Cyber-aggression_Grade6	1					
2. Cyber-aggression_Grade7	0.48 **	1				
3. Cyber-aggression_Grade8	0.36 **	0.67 **	1			
4. Cyber-aggression g_Grade9	NA	0.28 *	0.59 **	1		
5. Overt aggression T1	0.23 **	0.40 **	0.43 **	0.52 **	1	
6. Average hours online T1	0.60 **	0.33 **	0.33 **	0.22 *	0.02	1
M	0.23	0.30	0.38	0.40	0.21	2.83
*SD*	0.53	0.47	0.46	0.49	0.31	3.14

Note. For gender, girl was coded as 1, boy was coded as 0. * *p* < 0.05. ** *p* < 0.01.

**Table 2 ijerph-21-00429-t002:** Results of different growth mixture modeling analyses.

Class	AIC	BIC	ABIC	Entropy	LMR	BLRT (*p*)	Class Probability
1	866.33	900.99	872.45	N/A	N/A	N/A	1
2	630.77	677.00	638.93	0.92	45.3	<0.001	0.954/0.046
3	566.02	623.81	576.22	0.95	15.2	<0.001	0.857/0.093/0.049
4	481.96	551.30	494.19	0.93	10.1	<0.001	0.885/0.069/0.037/0.008

Note. *N* = 384; AIC, Akaike information criterion; BIC, Bayesian information criterion; ABIC, sample-size adjusted BIC; LMR, Lo–Mendell–Rubin likelihood ratio test; BLRT, bootstrap likelihood ratio test.

**Table 3 ijerph-21-00429-t003:** Associations between cyber-aggression trajectory class membership and gender, overt aggression, and time spent online at Time 1 using the three-step procedure.

Comparison (=1) Class 1 (High-Decreasing)	High-Decreasing	Low-Increasing
	b (SE)	b (SE)
2 (Low-increasing)		
Gender	0.30 (0.69)	-
Overt aggression	−1.40 (0.19) **	-
Average hours online	−1.71 (1.32) **	-
3 (Stable)		
Gender	−1.04 (1.37)	−0.1.34 (1.35)
Overt aggression	0.20 (0.14)	0.60 (0.17) **
Average hours online	2.16 (1.5)	1.86 (1.4) **

Note. All values are unstandardized; For gender, girl was coded as 1, boy was coded as 0; ** *p* < 0.01.

## Data Availability

The datasets generated during and/or analyzed during the current study are available from the corresponding author on reasonable request.

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
