# Peer review of "Developmental Trajectories of Cyber-Aggression among Early Adolescents in Canada: The Impact of Aggression, Gender, and Time Spent Online"

_ijerph, 2024, doi:10.3390/ijerph21040429_

Round 1
Reviewer 1 Report
Comments and Suggestions for Authors
Comments to the Authors:
The reviewer examined the submitted paper titled "Developmental Trajectories of Cyberbullying among Early Adolescents in Canada: The Impact of Aggression, Gender, and Time Spent Online" with great interest.
The submitted paper challenged to specify the predictors for cyberbullying among early adolescents in Canada by using the online panel data conducted by the authors. Consequently, the authors successfully clarified the significant effects of overt aggression and time spent online on cyberbullying. The reviewer believes that the submitted paper has the potential to make a significant contribution to the literature.
The reviewer confirms that the submitted paper carefully explored the association between overt aggression, time spent online, and cyberbullying using Latent Growth Modeling and found interesting findings. On the other hand, the reviewer has concerns about the authors' interpretation of these findings. Therefore, the reviewer believes that the submitted paper should be revised to address these concerns. After adequately revising the manuscript, the reviewer will recommend the submitted paper for publication.
Firstly, the authors did not discuss why cyberbullying sharply decreased from Grade 6 to Grade 9 among the students categorized into the high-decreasing group. Similarly, they did not discuss why the level of cyberbullying remained stable from Grade 6 to Grade 9 among the students categorized into the stable group. The reviewer believes that the authors should examine the differences in trajectories of cyberbullying among the latent groups and the mechanisms that generated such differences.
Then, the reviewer notes that the fact that the levels of cyberbullying of the high-decreasing group and the stable group seem to converge on a certain level might have significant meaning for understanding the differences between the latent groups.
Possibly, the convergence of the level of cyberbullying between the high-decreasing group and the stable group can be explained by "regression to the mean." If the convergence can be explained as a regression to the mean, it would mean that the differences between the high-decreasing group and the stable group are generated solely through differences in their initial states. In other words, the latent level of cyberbullying might not be substantially different between the high-decreasing group and the stable group. The reviewer believes this interpretation is consistent with the results presented by the authors.
Secondly, the authors state "Table 3 contains results from multinomial logistic regression for each of the four classes" (lines 320-321). However, the analytical results presented by the authors seem to be based on a three-class model. The reviewer suspects that this is a typo. If so, the authors should correct it.
Thank you for the opportunity to review the submitted paper. The reviewer would be grateful if his comments contribute to improving it.
Author Response
Reviewer #1:
Firstly, the authors did not discuss why cyberbullying sharply decreased from Grade 6 to Grade 9 among the students categorized into the high-decreasing group. Similarly, they did not discuss why the level of cyberbullying remained stable from Grade 6 to Grade 9 among the students categorized into the stable group. The reviewer believes that the authors should examine the differences in trajectories of cyberbullying among the latent groups and the mechanisms that generated such differences.
Response: Thank you for this suggestion. We have now added this; please see page 10.
Then, the reviewer notes that the fact that the levels of cyberbullying of the high-decreasing group and the stable group seem to converge on a certain level might have significant meaning for understanding the differences between the latent groups.
Response: Thank you for this suggestion. We have now added this on page 10.
Possibly, the convergence of the level of cyberbullying between the high-decreasing group and the stable group can be explained by "regression to the mean." If the convergence can be explained as a regression to the mean, it would mean that the differences between the high-decreasing group and the stable group are generated solely through differences in their initial states. In other words, the latent level of cyberbullying might not be substantially different between the high-decreasing group and the stable group. The reviewer believes this interpretation is consistent with the results presented by the authors.
Response: Thank you for this suggestion. We have added this as well on page 10.
Secondly, the authors state "Table 3 contains results from multinomial logistic regression for each of the four classes" (lines 320-321). However, the analytical results presented by the authors seem to be based on a three-class model. The reviewer suspects that this is a typo. If so, the authors should correct it.
Response: Thanks for bringing this to our attention. We have now revised this on page 9.
Reviewer 2 Report
Comments and Suggestions for Authors
The paper "Developmental Trajectories of Cyberbullying among Early Adolescents in Canada: The Impact of Aggression, Gender, and Time Spent Online" is a commendable manuscript based on longitudinal data. However, certain aspects need to be refined.
The part that defines cyberbullying - it would be important to look at the different aspects of the definition. Bullying implies that attacks happen at least 3-4 times a month and that there is a difference in power/strength. One should consider whether this is included in the behaviors that are measured or are cyber violence.
The authors emphasize the importance of cultural factors as a motivation to examine developmental trajectories in Canadian society. It would be useful to analyze what cultural differences they notice about previous studies of these phenomena and to indicate which variables in their research follow exactly these cultural differences.
Emphasizing Bandura's theory while emphasizing individual factors is a bit unusual. Much more emphasis is found in Bronfennrener's theory for the same. However, Bandura emphasized social perception and social cognition more than individual factors - the individual is an actor who observes the actions of others, concludes about their behavior and the consequences they experience, and then adjusts his actions to what he observes. What are the indicators, say, of gender differences in classic research?
Lykert type scale (for Cyberbullying scale) implies that participants express agreement or disagreement with the content of the items, and this is a frequency scale.
What is the meaning of expressing M and SD for gender?
M for cyberbullying is .23, and SD=.53 - therefore, they are not cyberbullies (if we take the 3xSD range of answers it says that they either never or behaved in the described ways rarely up to once a month) - what is the point of talking about cyberbullying? That's cyber violence. Since it is clear that the distributions are asymmetric, why was M calculated? What results conversions were made to perform the modeling?
When looking at Figure 1, it is evident that the High decreasing group are not truly cyberbullies - their highest average is between what happens monthly and what happens weekly, therefore, far less than 3-4 times a month which is used as the frequency of behavior in traditional bullying. The data is useful, but I think that the terms should be adjusted to reality.
In addition to modeling, it would be descriptive to calculate, for three subgroups, their average values for overt aggression and time online, and to show the shares of boys and girls in these three subgroups. Namely, when writing in the discussion about the role of time online, addiction to the Internet is mentioned - we have long since separated from time itself as an indicator of addiction, the motives, way of spending time, and consequences in the form of craving and giving up on other activities are more important. In the context of more time online as more exposure (it is the counterpart of a riskier and less risky environment in the physical world, if you go from the position of Bandura's theory - they can be more exposed to risky models, which the authors describe later in the section), the amount of time is relevant, but it is not necessarily related to internet addiction.
What are the results of gender differences in overt aggression? Namely, the authors mention that boys are more prone to physical aggression, and girls to relational aggression - what do the measures used measure? Are there cultural differences or differences in the measures used?
Author Response
The part that defines cyberbullying - it would be important to look at the different aspects of the definition. Bullying implies that attacks happen at least 3-4 times a month and that there is a difference in power/strength. One should consider whether this is included in the behaviors that are measured or are cyber violence.
Response: Thank you for bringing this to our attention. We agree that it's crucial to consider the traditional criteria of bullying, which typically involves repeated attacks (often defined as occurring 3-4 times a month), as well as a power imbalance. However, in the digital realm, the frequency and power dynamics can manifest differently. For instance, a single online act can have a widespread and lasting impact, challenging the traditional frequency criterion of bullying. Similarly, power imbalances may not be as straightforward as in face-to-face interactions and can be influenced by factors such as technological expertise or anonymity. Therefore, in assessing behaviors in the context of cyberbullying or cyber violence, it's essential to adapt these traditional bullying criteria to the nuances of the digital environment, considering how the nature of digital interactions might alter the manifestation of these elements. All off this said, we have now changed our language to better reflect that we are talking about aggression (cyber-aggression) and not necessarily bullying, as traditionally defined.
The authors emphasize the importance of cultural factors as a motivation to examine developmental trajectories in Canadian society. It would be useful to analyze what cultural differences they notice about previous studies of these phenomena and to indicate which variables in their research follow exactly these cultural differences.
Response: Thank you for your suggestion, and we have now revised our approach recognizing that we had not initially considered the potential for cultural differences to affect developmental trajectories in Canadian society. Consequently, while we initially expected cyberbullying trajectories among Canadian adolescents to mirror those observed in other countries, this deeper exploration into the Canadian context has provided more nuanced insights into the distinct ways cyberbullying manifests and operates within Canada's unique cultural and societal framework. Furthermore, by conducting a comparative analysis of cyberbullying trajectories in Canada, we contribute to the broader literature, enhancing our understanding by including a wider array of countries and populations. Please see page 3.
Emphasizing Bandura's theory while emphasizing individual factors is a bit unusual. Much more emphasis is found in Bronfennrener's theory for the same. However, Bandura emphasized social perception and social cognition more than individual factors - the individual is an actor who observes the actions of others, concludes about their behavior and the consequences they experience, and then adjusts his actions to what he observes. What are the indicators, say, of gender differences in classic research?
Response: Thanks for bringing this to our attention. We have now revised this on page 3.
Lykert type scale (for Cyberbullying scale) implies that participants express agreement or disagreement with the content of the items, and this is a frequency scale.
Response: Thanks for bringing this to our attention. We have now revised this (please see page 6).
What is the meaning of expressing M and SD for gender?
Response: We have now removed this. Please see page 8.
M for cyberbullying is .23, and SD=.53 - therefore, they are not cyberbullies (if we take the 3xSD range of answers it says that they either never or behaved in the described ways rarely up to once a month) - what is the point of talking about cyberbullying? That's cyber violence. Since it is clear that the distributions are asymmetric, why was M calculated? What results conversions were made to perform the modeling?
Response: As noted above, we have now changed our terminology to cyber-aggression to better reflect the construct we are measuring. Although the distributions are asymmetric, calculating the mean might seem counterintuitive as the mean can be skewed by outliers and may not accurately represent the central tendency of the data. However, we believed that the mean can still provide valuable information about the overall level of a variable within the sample. In addition, we employed Maximum Likelihood Robust (MLR) estimation in Mplus to address the asymmetry in the distribution.
When looking at Figure 1, it is evident that the High decreasing group are not truly cyberbullies - their highest average is between what happens monthly and what happens weekly, therefore, far less than 3-4 times a month which is used as the frequency of behavior in traditional bullying. The data is useful, but I think that the terms should be adjusted to reality.
Response: Thank you for this. As noted, we have now changed our terminology to cyber-aggression throughout the manuscript.
In addition to modeling, it would be descriptive to calculate, for three subgroups, their average values for overt aggression and time online, and to show the shares of boys and girls in these three subgroups. Namely, when writing in the discussion about the role of time online, addiction to the Internet is mentioned - we have long since separated from time itself as an indicator of addiction, the motives, way of spending time, and consequences in the form of craving and giving up on other activities are more important. In the context of more time online as more exposure (it is the counterpart of a riskier and less risky environment in the physical world, if you go from the position of Bandura's theory - they can be more exposed to risky models, which the authors describe later in the section), the amount of time is relevant, but it is not necessarily related to internet addiction.
Response: Thank you for highlighting this issue. We implemented a 3-step approach to incorporate covariates for predicting class membership, which complicates the calculation of average values for overt aggression and time spent online for each group. We have now updated our discussion to better reflect the time spent online; please see page 11.
What are the results of gender differences in overt aggression? Namely, the authors mention that boys are more prone to physical aggression, and girls to relational aggression - what do the measures used measure? Are there cultural differences or differences in the measures used?
Response: The focus on gender differences in overt aggression falls outside the scope of our current study. Our interest primarily lies in exploring the gender differences in cyberbullying trajectories. Regarding overt aggression, we did not assess relational aggression in this research. We have acknowledged this as a limitation in the relevant section of our study (please see page 12).